# Uteroplacental Insufficiency Causes Microbiota Disruption and Lung Development Impairment in Growth-Restricted Newborn Rats

**DOI:** 10.3390/nu14204388

**Published:** 2022-10-19

**Authors:** Yu-Chen S. H. Yang, Hsiu-Chu Chou, Yun-Ru Liu, Chung-Ming Chen

**Affiliations:** 1Joint Biobank, Office of Human Research, Taipei Medical University, Taipei 110301, Taiwan; 2Department of Anatomy and Cell Biology, School of Medicine, College of Medicine, Taipei Medical University, Taipei 110301, Taiwan; 3Department of Pediatrics, Taipei Medical University Hospital, Taipei 110301, Taiwan; 4Department of Pediatrics, School of Medicine, College of Medicine, Taipei Medical University, Taipei 110301, Taiwan; 5TMU Research Center for Digestive Medicine, Taipei Medical University, Taipei 110301, Taiwan

**Keywords:** uteroplacental insufficiency, intrauterine growth restriction, microbiota, radial alveolar count, vascular endothelial growth factor, von Willebrand factor

## Abstract

Preclinical studies have demonstrated that intrauterine growth retardation (IUGR) is associated with reduced lung development during the neonatal period and infancy. Uteroplacental insufficiency (UPI), affecting approximately 10% of human pregnancies, is the most common cause of IUGR. This study investigated the effects of UPI on lung development and the intestinal microbiota and correlations in newborn rats with IUGR, using bilateral uterine artery ligation to induce UPI. Maternal fecal samples were collected on postnatal day 0. On postnatal days 0 and 7, lung and intestinal microbiota samples were collected from the left lung and the lower gastrointestinal tract. The right lung was harvested for histological assessment and Western blot analysis. Results showed that UPI through bilateral uterine artery ligation did not alter the maternal gut microbiota. IUGR impaired lung development and angiogenesis in newborn rats. Moreover, on postnatal day 0, the presence of *Acinetobacter* and *Delftia* in the lungs and *Acinetobacter* and *Nevskia* in the gastrointestinal tract was negatively correlated with lung development. *Bacteroides* in the lungs and *Rodentibacter* and *Romboutsia* in the gastrointestinal tract were negatively correlated with lung development on day 7. UPI may have regulated lung development and angiogenesis through the modulation of the newborn rats’ intestinal and lung microbiota.

## 1. Introduction

Recent evidence indicates that adverse intrauterine environments detrimentally disrupt fetal lung development, resulting in persistent changes in lung structure and compromised respiratory function in postnatal life [1]. Preclinical studies, including our own, have demonstrated that intrauterine growth retardation (IUGR) is associated with impaired lung development during the neonatal period and infancy [2,3]. Human studies have established IUGR as a risk factor for decreased lung function in infancy, childhood, and adulthood [4,5,6,7,8]. Uteroplacental insufficiency (UPI), affecting approximately 10% of human pregnancies, is the most common cause of IUGR [9]. It is characterized by impaired placental function and compromised nutrient and oxygen delivery to the fetus [10]. A meta-analysis determined that placental dysfunction disrupts normal fetal growth and development and is strongly associated with bronchopulmonary dysplasia [11]. Overall, the evidence suggests that IUGR and subsequent small for gestational age status critically affects childhood outcomes. Thus, interventions that enhance lung development and function in these high-risk groups must be developed [12].

Microbiota are ecological communities of commensal, symbiotic, and pathogenic microorganisms found in and on all multicellular organisms from plants to animals [13]. The intestinal microbiota play crucial roles in maintaining health [14,15]. Preclinical studies have revealed that IUGR disrupts intestinal microbiota development during both the immediate postnatal period and adulthood [16,17,18]. However, the effects of UPI on the lung microbiota remains unclear, as do the relationships between IUGR, lung and intestinal microbiota, and lung development. We postulated that UPI would alter lung and intestinal microbiota and that such disruptions would influence the lung development of newborn rats with IUGR. Aside from exploring the effects of UPI on lung and intestinal microbiota, we assessed the correlations between intestinal and lung microbiota and lung development in newborn rats with IUGR.

## 2. Materials and Methods

### 2.1. Animal Model

Time-dated pregnant Sprague Dawley rats were housed individually in laboratory cages with a 12:12 light–dark cycle and were given ad libitum access to food and water. On gestation day 18, the IUGR group underwent UP, induced through bilateral uterine artery ligation. The control group underwent sham surgery [3]. Dams recovered quickly from the procedures and resumed feeding the same day. All rats were delivered naturally at term (22 days). The day of birth was designated postnatal day 0, and the litters were weighed. To ensure comparability and to prevent differing litter sizes from affecting the study outcomes, the litters in both groups were culled to nine pups. After the pups were euthanized on postnatal days 0 and 7, the litters were culled to nine and four pups, respectively.

### 2.2. Tissue Collection and Processing

On postnatal days 0 and 7, rats from each group were randomly selected for examination, irrespective of sex. The animals were euthanized with an overdose of isoflurane. Next, lung and intestinal microbiota samples were collected from the left lung and the lower gastrointestinal tract, and the right lung was harvested. Sampling was conducted under a culture-independent approach, specifically 16S rRNA-based microbial community profiling on the Illumina MiSeq platform (Illumina, San Diego, CA, USA). Prior to each harvest, the instruments were rinsed with ethanol and flamed. The murine lungs were excised, placed in tubes containing 1 mL of sterile water, and homogenized mechanically using the Tissue Tearor (BioSpec Products, Bartlesville, OK, USA). Between samplings, the tissue homogenizer was cleaned and rinsed with ethanol and water. Water control samples from the homogenization that had been exposed to clean instruments were sequenced as procedural controls. To assess the impact of UPI on the composition of the maternal gut microbiota, gut microbial profiling was performed on maternal fecal samples collected on postnatal day 0.

### 2.3. Lung Histology

The right middle lobes of the lungs were harvested and then dehydrated in alcohol, cleared in xylene, and embedded in paraffin. Five micrometer tissue sections were stained with hematoxylin and eosin. Lung and intestinal morphometry examinations proceeded under light microscopy. The radial alveolar count (RAC) was determined to evaluate the structural development of alveoli by using a method modified from that of Cooney et al., [19]. In brief, a perpendicular line was drawn from the center of a respiratory bronchiole to the edge of the pleura and the number of alveoli cut by this line was counted manually.

### 2.4. Immunohistochemistry of Vascular Endothelial Growth Factor and von Willebrand Factor

Immunohistochemical staining was performed on 5-μm paraffin sections. After the sections were deparaffinized, heat-induced epitope retrieval was performed by immersing the slides in 0.01 M sodium citrate buffer (pH 6.0). To block endogenous peroxidase activity and the nonspecific binding of antibodies, the sections were preincubated for 1 h at room temperature in 0.1 M phosphate-buffered saline containing 10% normal goat serum and 0.3% H2O2. The sections were incubated with one of two primary antibodies: rabbit polyclonal anti-von Willebrand factor (vWF) antibody (1:100; Abcam, Cambridge, MA, USA) or rabbit polyclonal anti-vascular endothelial factor (VEGF) antibody (1:50; Santa Cruz Biotechnology, Inc., Santa Cruz, CA, USA). The sections were treated with biotinylated goat antirabbit immunoglobulin G (IgG; 1:200, Jackson ImmunoResearch Laboratories, Inc., West Grove, PA, USA). An avidin–biotin complex kit (Vector Laboratories, Inc., Burlingame, CA, USA) and a diaminobenzidine substrate kit (Vector Laboratories, Inc.) were used to visualize the brown reaction product, following the manufacturer’s recommendations. Vascular density was determined through immunohistochemical staining for vWF. Microvessel density was determined by counting the number of vessels positive for vWF staining in at least four random lung fields at 400× magnification in an unbiased manner [20].

### 2.5. Western Blot Analysis of VEGF

Lung tissues were homogenized in ice cold buffer containing 50 mM Tris·HCl (pH 7.5), 1 mM egtazic acid, 1 mM ethylenediaminetetraacetic acid, and a protease inhibitor cocktail (complete minitablets; Roche, Mannheim, Germany). The samples were sonicated and then centrifuged at 500× *g* for 20 min at 4 °C to remove cellular debris. Proteins (30 μg) were resolved on 12% sodium dodecyl sulfate–polyacrylamide gel electrophoresis gels under reducing conditions. Next, they were electroblotted onto a polyvinylidene difluoride membrane (ImmobilonP, Millipore, Bedford, MA, USA). Following blocking with 5% nonfat dry milk, the membranes were incubated with an anti-VEGF antibody or an anti-β-actin antibody (Sigma-Aldrich, St. Louis, MO, USA) and subsequently with horseradish peroxidase-conjugated goat antirabbit IgG or antimouse IgG (Pierce Biotechnology, Rockford, IL, USA). Protein bands were detected using SuperSignal Substrate (Pierce Biotechnology). Densitometric analysis was performed to measure the intensity of VEGF and β-actin bands using AIDA software 8.20.

### 2.6. 16S rRNA Gene Sequencing and Analysis

Herein, 16S rRNA analysis was conducted as described in a 2020 article by Yang et al., [21]. Using the QIAamp Fast DNA Stool Mini Kit and QIAamp DNA Microbiome Kit (QIAGEN, Hilden, Germany), the microbial DNA was purified from stool and lung tissue samples, respectively. The library preparation followed the protocol of 16S rRNA gene amplicon sequencing on the Illumina MiSeq System (Illumina, San Diego, CA, USA). The gene-specific sequences used, which target the 16S V3 and V4 regions, were removed from the demultiplexed paired reads by using the Cutadapt tool (Version 1.12 DOI: https://doi.org/10.14806/ej.17.1.200). Following the workflow described by Callahan et al., in their 2016 paper, the filtered reads were processed in the R environment (Version 3.6.1) using R package DADA2 (Version 1.14.1 DOI https://doi.org/10.1038/nmeth.3869) [22]. Taxonomy assignment was performed using the SILVA database (Version 138) [23] as the reference, with a minimum bootstrap confidence of 80. Multiple sequence alignment of the structural variations was performed with DECIPHER (Version 2.14.0), and a phylogenetic tree was constructed from the alignment by using the phangorn package (Version 2.5.5) [24]. The count table, taxonomy assignment results, and phylogenetic tree were consolidated into a phyloseq object, and community analyses were performed using phyloseq (Version 1.30.0) [25]. The α-diversity indices were calculated using the estimate richness function from the phyloseq package. Statistical comparison between the IUGR and the control groups was conducted through the Kruskal–Wallis and Wilcoxon tests (α = 0.05). UniFrac distances were calculated using the GUniFrac package (Version 1.1) to assess group community dissimilarity [26]. Principal coordinate analysis (PCoA) ordination on UniFrac distances was performed. The adonis function from the vegan package (Version 2.5.6) was employed to statistically analyze the between group differences in composition, whereas the betadisper package was used to examine the homogeneity of dispersion.

### 2.7. Statistical Analysis

Alpha diversity was evaluated using the inverse Simpson index. Beta diversity was estimated by computing unweighted UniFrac distances, which were used as inputs for PCoA. Differences in alpha diversity and relative taxon abundance were evaluated over time through the Wilcoxon group, both within and between study groups. The significance of the separation between study groups on the PCoA plot was tested through permutational multivariate analysis of variance by using the adonis function in the vegan package. Data are shown as box and whisker plots in the microbiota or presented as means ± standard deviations. Correlations between the relative abundances of bacterial genera and the RAC were analyzed using Spearman’s rank correlation test. Differences were considered statistically significant at *p* < 0.05.

## 3. Results

In total, 39 control pups were delivered by three sham-operated dams, and 26 IUGR rats were delivered by four dams in which UPI was induced. The mean birth weight of the IUGR rats (5.89 ± 0.74 g) was significantly lower than that of the control rats (6.36 ± 0.55 g; *p* < 0.01). On postnatal day 0, we retrieved 12 and 8 pups from the sham-operated and UPI-induced dams, respectively, for examination. On postnatal day 7, we retrieved eight and eight pups from the sham-operated and UPI-induced dams, respectively. Compared with the control rats, the IUGR rats had significantly lower body weights on both postnatal day 0 (5.38 ± 0.87 g versus 6.40 ± 0.56 g; *p* < 0.01) and postnatal day 7 (14.50 ± 0.55 g versus 16.90 ± 1.39 g; *p* < 0.001).

### 3.1. UPI Impaired Lung Development in Growth Restricted Newborn Rats

Images of representative lung sections stained with hematoxylin and eosin are shown in Figure 1A. We measured the RAC as an index of the alveolar number. The lungs of rats born to control dams exhibited normal morphology on postnatal days 0 and 7. On both those days, the lungs of the IUGR rats exhibited a significantly lower RAC than the control rats (Figure 1B).

### 3.2. UPI Impaired Lung Angiogenesis in Growth-Restricted Newborn Rats 

Representative immunohistochemical images and Western blots of VEGF are presented in Figure 2, as are representative immunohistochemical images for vWF and the results of semiquantitative analysis for vascular density. The control rats exhibited similar VEGF and vWF immunoreactivity on postnatal days 0 and 7. On postnatal days 0 and 7, the VEGF expression, vWF immunoreactivity, and vascular density of the IUGR rats were significantly lower than those of the control group (Figure 2A,B, *p* < 0.001).

### 3.3. UPI Did Not Alter the Maternal Intestinal Microbiota

Figure 3 presents the impact of bilateral uterine artery ligation (IUGR group) and sham surgery (control group) on gestation day 18 on the maternal gut microbiota. The procedure did not alter α-diversity indices (observed, Chao1 index, Shannon, or Simpson), β-diversity, or bacterial composition at the phylum level of the maternal gut microbiota. The heatmap of the gut microbiota at the phylum level is shown in Appendix A.

### 3.4. IUGR Altered the Lung and Intestinal Microbiota on Postnatal Day 0

As shown in Figure 4A,D, the lung and intestinal microbiota of the IUGR group exhibited significantly lower α-diversity indices than the control group on postnatal day 0. The microbial community structure of each group was assessed through PCoA of unweighted pairwise UniFrac distances. Overall, the microbiota in the IUGR group differed significantly from those in the control group (Figure 4B,E).

The lung microbiota in the two groups contained four major bacterial phyla, accounting for more than 97% of the total sequences (Figure 4C and Appendix A). Specifically, in the control and IUGR groups, these were Bacteroidota (55.4 ± 0.0% and 50.2 ± 0.1%, respectively); Firmicutes (30.1 ± 0.0% and 23.7 ± 0.0%, respectively); Fusobacteriota (4.7 ± 0.0% and 2.2 ± 0.0%, respectively); Proteobacteria (7.0 ± 0.0% and 21.0 ± 0.1%, respectively). IUGR significantly reduced the relative abundance of Bacteroidota, Firmicutes, and Fusobacteriota relative to those in the control group; moreover, it significantly increased the relative abundance of Proteobacteria relative to the control group. The intestinal microbiota in the two groups contained four major bacterial phyla, accounting for more than 97% of the total sequences. Specifically, in the control and IUGR groups, these were Bacteroidota (33.7 ± 0.2% and 33.3 ± 0.2%, respectively); Firmicutes (28.9 ± 0.1% and 25.6 ± 0.1%, respectively); Fusobacteriota (3.3 ± 0.0% and 0.9 ± 0.0%, respectively); Proteobacteria (31.3 ± 0.2% and 36.6 ± 0.2%, respectively). The relative abundance of Fusobacteriota was significantly lower in the IUGR group than in the control group (Figure 4F and Appendix A).

### 3.5. IUGR Altered the Lung Microbiota but Did Not Alter the Intestinal Microbiota on Postnatal Day 7

Regarding the lung microbiota, as shown in Figure 5A,B, the IUGR group exhibited significantly lower α-diversity and different β-diversity than the control group on postnatal day 7. By contrast, the α-diversity indices or β-diversity relating to the intestinal microbiota did not differ between the groups on postnatal day 7 (Figure 5D,E). On that day, the phylum-level bacterial composition of the intestine and lung microbiota was similar between the two groups (Figure 5C,F and Appendix A).

### 3.6. Lung Microbiota Compositions on Postnatal Days 0 and 7

The total relative abundances of the 10 most abundant genera in the control and IUGR groups on postnatal day 0 were 65.9% and 72.0%, respectively. *Acinetobacter*, *Nevskia*, and *Delftia* were significantly enriched in the IUGR group, whereas *Prevotella*, *Fusobacterium*, and *Finegoldia* were significantly enriched in the control group (Figure 6A). The total relative abundance of the 10 most abundant genera in the control and the IUGR groups on postnatal day 7 were 65.7% and 69.3%, respectively. *Bacteroides*, *Sut-terella*, and *Alloprevotella* were significantly enriched in the IUGR group, whereas *Prevotella* and *Fusobacterium* were significantly enriched in the control group (Figure 6B).

Moreover, *Acinetobacter* and *Delftia* in the lungs and *Acinetobacter* and *Nevskia* in the gastrointestinal tract were significantly negatively correlated with the RAC (Figure 7A, *p* < 0.01) on day 0. *Bacteroides* in the lungs and *Rodentibacter* and *Romboutsia* in the gastrointestinal tract were significantly negatively correlated with the RAC (Figure 7B, *p* < 0.01) on day 7.

## 4. Discussion

Our in vivo model demonstrated that UPI induced IUGR, as indicated by the reduced weight of the pups at birth and on postnatal day 7. However, UPI through bilateral uterine artery ligation did not alter the maternal gut microbiota. IUGR impaired lung development in the newborn rats, altering the lung microbiota on postnatal days 0 and 7 and altering the intestinal microbiota on postnatal day 0. UPI may have impaired lung development through the modulation of the lung and intestinal microbiota in the newborn rats. Improving fetal nutrition by optimizing maternal nutrition is recommended for advancing lung development in growth-restricted newborns.

Interestingly, *Acinetobacter* was negatively correlated with lung development and existed in the lungs and gastrointestinal tract in our animal model. *Acinetobacter* infection in children is a serious disease and may causing prolonged disabilities and death [27]. Moreover, a case reporting shown *Delftia* may correlated with sepsis in an immune deficient child [28]. *Delftia* acidovorans may infect immunocompetent and immunocompromised patients [29]. In addition, *Bacteroides* detected in the lungs may correlate with sepsis and acute respiratory distress syndrome [30]. However, in our UPI-induced IUGR model, these bacteria may be from the maternal animal and their changes are negatively correlated with lung development, and the mechanisms of *Acinetobacter*, *Delftia*, and *Bacteroides* and how they may affect lung development are unclear.

Early life changes in the intestinal microbiota lead to alterations in gut physiology, which can exert lifelong effects on general health [31]. The effects of IUGR on the intestinal microbiota have been reported in only one rat model. Fança-Berthon et al., found that IUGR induced by isocaloric protein restriction during pregnancy increased bacterial density on postnatal day 5 and reduced bacterial density on postnatal day 12 in the offspring [16]. In this study, we observed that UPI altered the intestinal microbiota on postnatal day 0 and altered the lung microbiota on postnatal days 0 and 7. Furthermore, impaired lung development was noted on postnatal days 0 and 7. We postulate that the nutritional status alters the composition and diversity of lung and intestinal microbiota. The impaired lung development in the offspring correlates with the relative abundances of bacterial genera in the lung and intestinal microbiota. These results suggest that alterations in the lung and intestinal microbiota contribute to IUGR-induced lung development impairment in newborn rats. Such impairment may be mitigated through the manipulation of these microbiota.

Findings from animal models have suggested that maternal bacteria actively colonize the fetal gut [32,33]. Turenen et al., reported that the first-pass meconium of human neonates harbored a microbiome that may be explained by perinatal colonization or intrauterine colonization through bacterial extracellular vesicles [34]. These findings indicate that intrauterine nutrition is strongly linked to alterations in the lung and intestinal microbiota at birth of the rats in the present study.

Although the connection between prenatal nutrition and lung health has been described, the underlying mechanisms remain poorly understood. VEGF is a potent endothelial cell mitogen that is crucial for vasculogenesis and angiogenesis in the cardiovascular system during embryonic development [34]. Angiogenesis is essential for alveolarization, which in turn is a vital component of normal lung development [35]. In a murine model, maternal protein restriction reduced VEGF expression and impaired lung development in intrauterine growth-restricted fetuses on gestation day 20 [36]. Previously, the effects of UPI on VEGF expression and vascular development in growth-restricted offspring were unknown. Here, we demonstrated that uteroplacental insufficiency induced through bilateral uterine artery ligation on gestation day 18 reduced VEGF expression and vascular density in rat lungs on postnatal days 0 and 7. These results suggested that nutrition plays a crucial role on VEGF signaling in fetal lung development.

## 5. Conclusions

We observed that UPI through bilateral uterine artery ligation induced IUGR and disrupted lung microbiota on postnatal days 0 and 7 and altered intestinal microbiota on postnatal day 0 only. These findings indicate that intrauterine malnutrition may induce an imbalance of the intestinal and lung microbial community and impair lung development in the offspring and suggest that microbiota may be involved in the mechanism of IUGR-associated lung development impairment. Our findings demonstrate a significant cross-talk among lung and intestinal microbiota and lung development in growth-restricted newborn rats. Future studies are necessary to evaluate the effects of nutritional supplements on microbiota and lung development of the offspring and to increase fetal growth.

## Figures and Tables

**Figure 1 nutrients-14-04388-f001:**
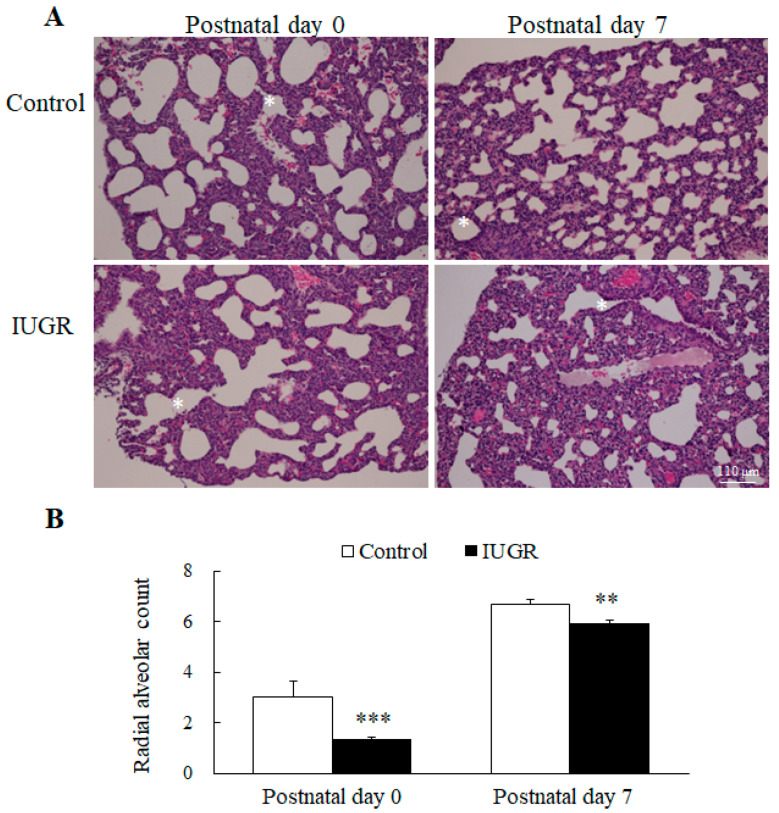
(**A**) Representative histological images of hematoxylin-and-eosin-stained murine lungs and (**B**) The radial alveolar count (RAC) of the lungs of control and intrauterine growth retardation (IUGR) rats on postnatal days 0 and 7. * denotes a respiratory bronchiole. Compared with the control rats, the IUGR rats exhibited a significantly lower RAC on these 2 days. *n* = 8–12 rats at each postnatal day. Data are presented as means ± standard deviations. ** *p* < 0.01, *** *p* < 0.001.

**Figure 2 nutrients-14-04388-f002:**
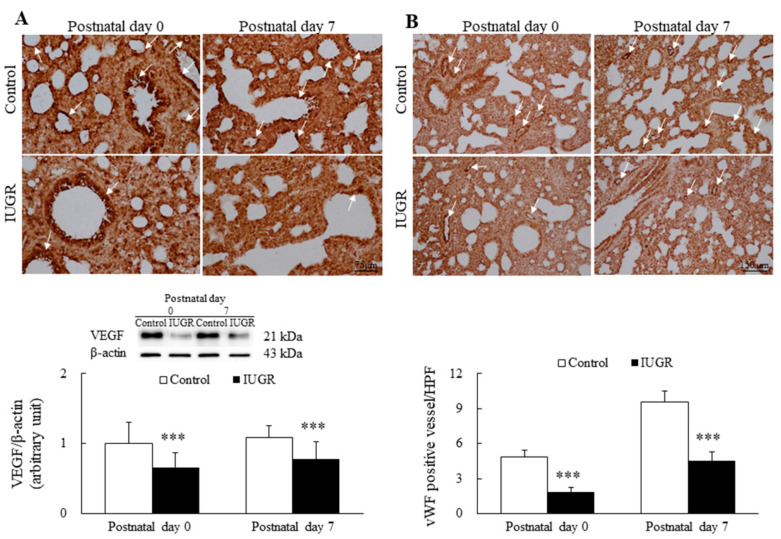
(**A**) Representative immunohistochemical images and Western blots of vascular endothelial factor (VEGF) and (**B**) representative immunohistochemical images for von Willebrand factor (vWF) and the results of semiquantitative vascular density analysis of control and IUGR rats on postnatal days 0 and 7. VEGF immunoreactivities (white arrow) were primarily detected in the endothelial and epithelial cells. Control rats exhibited similar VEGF and vWF immunoreactivities on postnatal days 0 and 7. By contrast, the VEGF expression, vWF immunoreactivity, and vascular density (white arrow) of the IUGR group were significantly lower than those of the control group on postnatal days 0 and 7. *n* = 7 rats at each postnatal day. Data are presented as means ± standard deviations. *** *p* < 0.001.

**Figure 3 nutrients-14-04388-f003:**
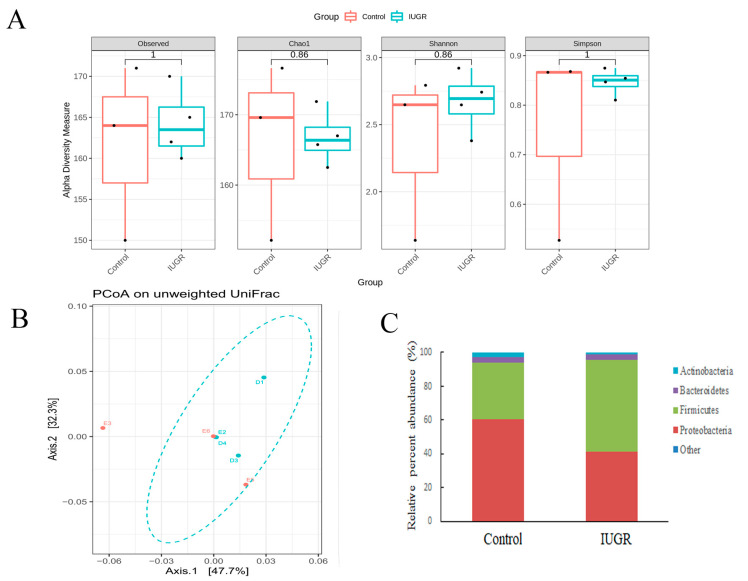
Impact of bilateral uterine artery ligation (IUGR group) or sham surgery (control group) on gestation day 18 on the maternal gut microbiota. Bilateral uterine artery ligation did not alter (**A**) α-diversity indices (observed, Chao1 index, Shannon, or Simpson), (**B**) β-diversity (PCoA on unweighted UniFrac distances), or (**C**) bacterial composition of the maternal gut microbiota at the phylum level. Other means the bacterial with lower abundances include Acidobacteriota, Campilobac-terota, Cyanobacteria, Desulfobacterota, Fusobacteriota, Verrucomicrobiota. Data are shown as box-and-whisker plots. Control group *n* = 3 and IUGR group *n* = 4 dams in each group.

**Figure 4 nutrients-14-04388-f004:**
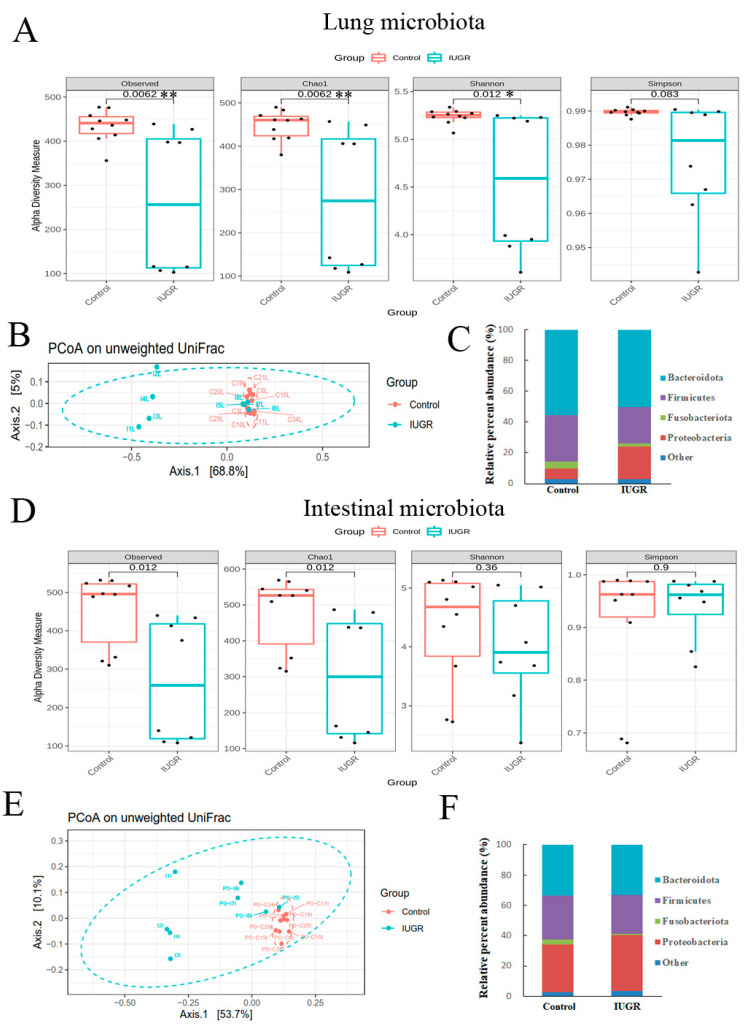
Effect of IUGR on the bacterial phase of offspring rats on postnatal day 0. The α-diversity indices (observed, Chao1 index, Shannon, and Simpson) in the (**A**) lungs or (**D**) gastrointestinal tract. The β-diversity (PCoA on unweighted UniFrac distances) of the (**B**) lungs or (**E**) intestinal tract and the bacterial composition at the phylum level in the (**C**) lungs or (**F**) intestinal tract. Compared with those of the control group, the lung and intestinal microbiota of the IUGR group exhibited significantly lower α-diversity and different β-diversity on postnatal day 0. *n* = 8–10 rats in each group. Data are shown as box-and-whisker plots. * *p* < 0.05, ** *p* < 0.01.

**Figure 5 nutrients-14-04388-f005:**
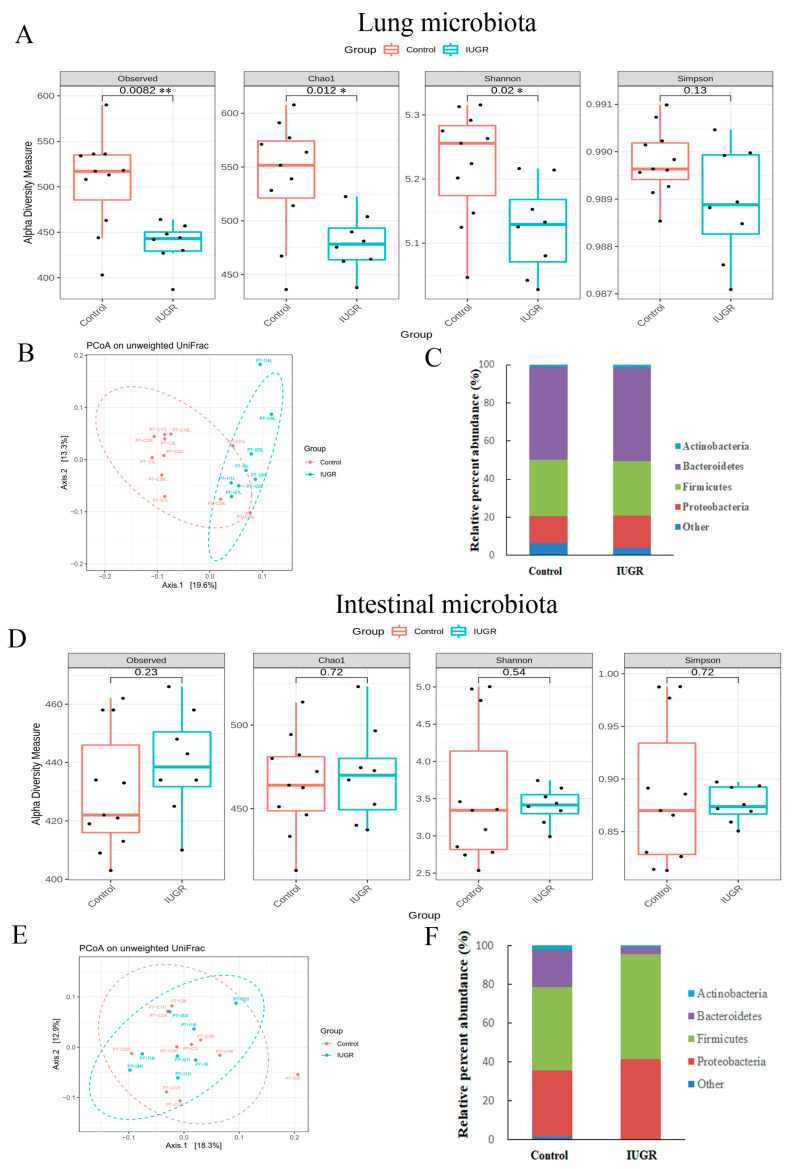
Effect of IUGR on the bacterial phase of offspring rats on postnatal day 7. The α-diversity indices (observed, Chao1 index, Shannon, and Simpson) in the (**A**) lungs or (**D**) gastrointestinal tract. The β-diversity (PCoA on unweighted UniFrac distances) of the (**B**) lungs or (**E**) intestinal tract and the bacterial composition at the phylum level in the (**C**) lungs or (**F**) intestinal tract. Compared with those of the control group, the lung and intestinal microbiota of the IUGR group exhibited significantly lower α-diversity and different β-diversity on postnatal day 7. The α-diversity and β-diversity of the intestinal microbiota did not differ significantly between the two groups. *n* = 8–10 rats in each group. Data are shown as box-and-whisker plots. * *p* < 0.05, ** *p* < 0.01.

**Figure 6 nutrients-14-04388-f006:**
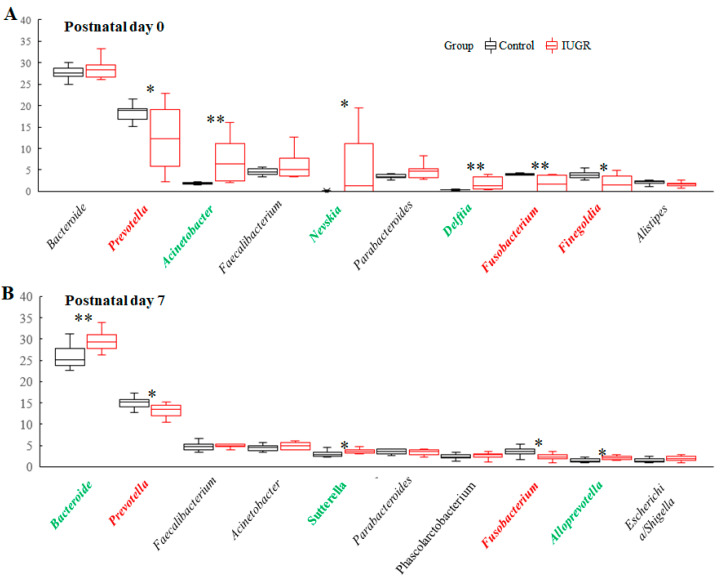
Relative abundances of the 10 most abundant genera in the lung microbiota of the control and IUGR groups on (**A**) postnatal day 0 and (**B**) postnatal day 7. On postnatal day 0, *Acinetobacter*, *Nevskia*, and *Delftia* were significantly enriched in the IUGR group, whereas *Prevotella*, *Fusobacterium*, and *Finegoldia* were significantly enriched in the control group. On postnatal day 7, *Bacteroides*, *Sutterella*, and *Alloprevotella* were significantly enriched in the IUGR group, whereas *Prevotella* and *Fusobacterium* were significantly enriched in the control group. Genera significantly enriched in the control and IUGR groups are labeled in red and green, respectively. *n* = 8–10 rats at each postnatal day. Data are shown as box-and-whisker plots. * *p* < 0.05, ** *p* < 0.01.

**Figure 7 nutrients-14-04388-f007:**
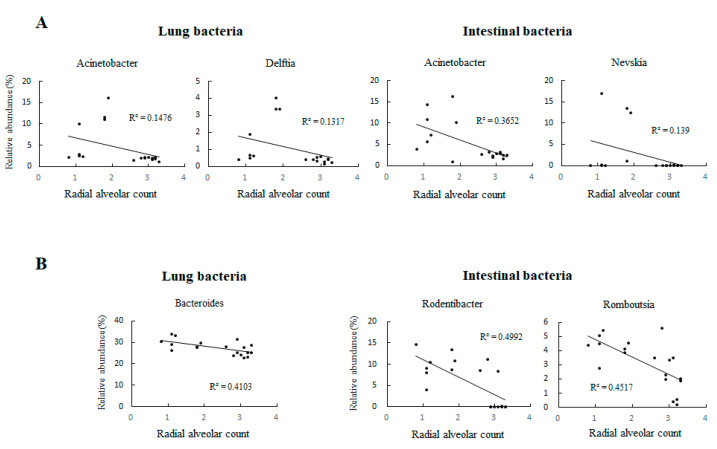
Correlations between lung and intestinal bacteria for which significant results were found at the genus level and the RAC on (**A**) postnatal day 0 and (**B**) postnatal day 7. On postnatal day 0, *Acinetobacter* and *Delftia* in the lungs and *Acinetobacter* and *Nevskia* in the gastrointestinal tract were significantly negatively correlated with the RAC (*p* < 0.01). On postnatal day 7, *Bacteroides* in the lungs and *Rodentibacter* and *Romboutsia* in the gastrointestinal tract were significantly negatively correlated with the RAC (*p* < 0.01). *n* = 8–10 rats at each postnatal day.

## Data Availability

The raw abundance table is provided in Appendix A.

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
