# Peer review of "Uteroplacental Insufficiency Causes Microbiota Disruption and Lung Development Impairment in Growth-Restricted Newborn Rats"

_nutrients, 2022, doi:10.3390/nu14204388_

Round 1

Reviewer 1 Report

The author observed that intrauterine growth retardation can affect the composition of lung and intestinal microbial communities, and found changes at different time points, which is an interesting discovery. However, there are still questions to be answered. First of all, is the change in the microbial community directly caused by IUGR or indirectly? In the current work, there is no evidence of this problem. In addition, some statistical conclusions in the study are not supported by data, which is not rigorous enough. The details are as follows:

Major:

Fig 1B: The n number should be marked in all statistical results.

Line 208-209: About vascular density, conclusive statements should be supported by statistical data.

Fig 3A: The number of groups is too small so the error value is too large.

Fig 4: Why are there two distinct groups between the repetitions in IUGR group? Whether the successful establishment of the model has been verified?

Fig 6: Why are different bacterial populations affected differently in the IUGR group? The author should explain this.

Minor:

Line 49-50: The statement is a little repetitive.

Fig 2A: The different lines of WB should be marked with the sample names.

Fig 6: The meanings of different colors of group names should be directly expressed in Figure Legend.

Author Response

Author Response to Reviewer 1 Comments:

Comments and Suggestions for Authors

The author observed that intrauterine growth retardation can affect the composition of lung and intestinal microbial communities, and found changes at different time points, which is an interesting discovery. However, there are still questions to be answered. First of all, is the change in the microbial community directly caused by IUGR or indirectly? In the current work, there is no evidence of this problem. In addition, some statistical conclusions in the study are not supported by data, which is not rigorous enough. The details are as follows:

Response: Thanks for reviewer’s comments. We have answered to the comments in a point by point reply and extensively corrected the text according to the reviewer’s comments.

Major:

Fig 1B: The n number should be marked in all statistical results.

Response: We added the number of animals in the revised figure legends.

Line 208-209: About vascular density, conclusive statements should be supported by statistical data.

Response: We corrected the sentences to “The VEGF expression, vWF immunoreactivity, and vascular density of the IUGR rats were significantly lower than those of the control group (Figure 2A and 2B, p < 0.001)” on line 202.

Fig 3A: The number of groups is too small so the error value is too large.

Response: Thanks for reviewer’s comments. Totally 39 control pups were delivered by three sham operated dams, and 26 IUGR rats were delivered by four dams in which UPI was induced. Statistical analysis revealed that UPI did not alter the maternal intestinal microbiota.

Fig 4: Why are there two distinct groups between the repetitions in IUGR group? Whether the successful establishment of the model has been verified?

Response: In total, 39 control pups were delivered by four sham operated dams, and 26 IUGR rats were delivered by three dams in which UPI was induced. The mean birth weight of the IUGR rats (5.89 ± 0.74 g) was significantly lower than that of the control rats (6.36 ± 0.55 g; p < 0.01). These results indicate that IUGR was established.

Fig 6: Why are different bacterial populations affected differently in the IUGR group? The author should explain this.

Response: Thanks for reviewer’s comments, previous studies have revealed that IUGR disrupts intestinal microbiota development during both the immediate postnatal period and adulthood [16-18]. However, how IUGR affect lung and intestinal microbiota, and lung development are remains unclear. This depiction is presented on page 2, lines 51-53 (Preclinical studies have revealed that IUGR disrupts intestinal microbiota development during both the immediate postnatal period and adulthood [16-18]) and page 11, lines 326-330 (The effects of IUGR on the intestinal microbiota have been reported in only one rat model. Fança-Berthon et al. found that IUGR induced by isocaloric protein restriction during pregnancy increased bacterial density on postnatal day 5 and reduced bacterial density on postnatal day 12 in the offspring [16]).

Minor:

Line 49-50: The statement is a little repetitive.

Response: Thanks for reviewer's comments, we have deleted the sentence “A healthy gut microbiota is essential for human health” in the revised manuscript.

Fig 2A: The different lines of WB should be marked with the sample names.

Response: The different lines of WB were marked with the sample names in the revised version.

Fig 6: The meanings of different colors of group names should be directly expressed in Figure Legend.

Response: The meanings of different colors of bacterial names were described as “Genera significantly enriched in the control and IUGR groups are labeled in red and green, respectively” in Figure legend 6.

Reviewer 2 Report

Line 126 : 16S rDNA must be 16S rRNA

Line 128:  “16S rDNA” was purified, it is not common to use this term, simply all the DNA was extracted , to be more  specific, the authors may say “microbial DNA was purified”

Line 155-  why did the authors used unweighted UniFrac distances  for Beta diversity? Although this index use the phylogenic relation, but it does not consider the abundance “unweighted” , weighted UniFrac might be used with other indices like Jaccard and Bray Curtis

 Line 226-230: figure 3, why the focus was on phyla rather than genus and species ? was the analysis of alpha diversity done using date derived from phyla or other taxonomic levels ? usually it is done from genus , or species if available , it will make a difference in the results, as phyla is not discriminative enough. For beta diversity ; please check the comment above  

Line 239 and figures : The phylum Bacteroidota is synonym for Bacteroidetes, but it is better to use either one in the entire paper, some figures used Bacteroidetes, the latter is more preferable than Bacteroidota

Line 279: from section 3.6: authors started discussing the genus , while all the previous sections were all about phyla, it is better to discuss all the taxonomic levels in the same section ,and the related figures, also , why the focus is on the top 10 genera ? species were not mentioned although they are present in the supplementary file, although it is known that illumine sequencing using V3-V4 region will not provide detailed information for the species which is a known limitation if short read sequencing

-results of linear discriminant analysis (LDA) were not mentioned in the results section with no supportive figures

- correlation analysis, did the authors try to correlate lung and intestinal bacteria in the offspring, also with those of the mothers ?

-heatmaps can be generated , they can show differentially abundant microbiota and clearly show the difference between groups

Line 320: discussion : is rather short, it can be improved if the results section was improved as advised above, it is important to explain the connection between types of bacteria present or absent, enriched or depleted and effect on lung development and gut as well.

Line 360:intestinal and lung dysbiosis” can you justify this conclusion ?

General comments :

1-      names of bacteria are better to be italicized, from phyla downwards

2-      improve the resolution of the figures

Author Response

Author Response to Reviewer 2 Comments:

Line 126: 16S rDNA must be 16S rRNA

Response: We have corrected “16S rDNA” to “16S rRNA” in the revise manuscript.

Line 128: “16S rDNA” was purified, it is not common to use this term, simply all the DNA was extracted, to be more specific, the authors may say “microbial DNA was purified”

Response: Thanks for reviewer’s comments, we have corrected it as “the microbial DNA was purified from stool and lung tissue samples” in the revise manuscript.

Line 155- why did the authors used unweighted UniFrac distances for Beta diversity? Although this index use the phylogenic relation, but it does not consider the abundance “unweighted” , weighted UniFrac might be used with other indices like Jaccard and Bray Curtis

Response: Thanks for reviewer’s comments, unweighted UniFrac can be understood intuitively as the ratio of a phylogenetic tree to be exclusively dominated by a single cluster. The higher the ratio of exclusive branch length, the more distant the relationship between the two cluster members, and the greater the difference between the clusters.

Line 226-230: figure 3, why the focus was on phyla rather than genus and species? was the analysis of alpha diversity done using date derived from phyla or other taxonomic levels? usually it is done from genus, or species if available, it will make a difference in the results, as phyla is not discriminative enough. For beta diversity; please check the comment above 

Response: Thanks for reviewer’s comments. Alpha diversity is defined as the mean diversity of species of the sample, not for any level. Moreover, a statistically significant change in the Bacteridota content of phylum was found, so we focused on the phylum level.

Line 239 and figures: The phylum Bacteroidota is synonym for Bacteroidetes, but it is better to use either one in the entire paper, some figures used Bacteroidetes, the latter is more preferable than Bacteroidota

Response: Thanks for reviewer's comments, we used Bacteroidota in this article, and in the figure is Genus Bacteroides not Bacteroidetes.

Line 279: from section 3.6: authors started discussing the genus, while all the previous sections were all about phyla, it is better to discuss all the taxonomic levels in the same section ,and the related figures, also, why the focus is on the top 10 genera? species were not mentioned although they are present in the supplementary file, although it is known that illumine sequencing using V3-V4 region will not provide detailed information for the species which is a known limitation if short read sequencing

Response: Thanks for reviewer's comments, we have reassembled the narrative of section 3.6. Moreover, on postnatal day 0, the total relative abundance of the 10 most abundant genera in the control and IUGR groups was 65.9% and 72.0%, respectively. And the total relative abundances of the 10 most abundant genera in the control and IUGR groups at postnatal day 7 were 65.7% and 69.3%. We believe that this percentage represents most of the characteristics. We also agree NGS cannot provide detailed information for the species.

-results of linear discriminant analysis (LDA) were not mentioned in the results section with no supportive figures

Response: Thanks for reviewer's comments, we removed the description of the methods LDA analysis in the revise manuscript.

- correlation analysis, did the authors try to correlate lung and intestinal bacteria in the offspring, also with those of the mothers?

Response: Thanks for reviewer’s comments. The effects of UPI on the relationships among IUGR, lung and intestinal microbiota, and lung development remains unclear. The correlation analysis in Figure 7 are to assess the correlations between intestinal and lung microbiota and lung development in newborn rats with IUGR.

-heatmaps can be generated, they can show differentially abundant microbiota and clearly show the difference between groups

Response: Thanks for reviewer’s suggestions, we have added the heatmaps as Supplementary Figures 1-5 in the revised manuscript.

Line 320: discussion: is rather short, it can be improved if the results section was improved as advised above, it is important to explain the connection between types of bacteria present or absent, enriched or depleted and effect on lung development and gut as well.

Response: Thanks for reviewer’s comments, we added “Moreover, a case reported shown Delftia may correlated with sepsis in an immune deficient child [28]. Delftia acidovorans may infect the patients immunocompetent and immunocompromised [29]. In addition, Bacteroides were detected in lung may correlated with sepsis and the acute respiratory distress syndrome [30]. However, in our UPI-induced IUGR model, these bacteria may from maternal and their changes are negatively correlated with lung development, but the mechanisms of Acinetobacter, Delftia, and Bacteroides affect lung development are unclear” in Discussion on lines 318-324.

Line 360: “intestinal and lung dysbiosis” can you justify this conclusion?

Response: Thanks for reviewer’s comments. Dysbiosis is defined as an imbalance in the gut microbial community that is associated with disease. In this study, IUGR rats exhibited alterations in the relative abundance of microbes. We corrected the sentence to “These findings indicate that intrauterine malnutrition may induce imbalance of the intestinal and lung microbial community” in the revised manuscript.

General comments:

1- names of bacteria are better to be italicized, from phyla downwards

Response: Thanks for reviewer’s comments, we have corrected them in the revised manuscript.

2- improve the resolution of the figures

Response: Thanks for reviewer’s comments, we have improved the resolution of the figures in the revise manuscript.

Round 2

Reviewer 2 Report

Thanks for making all the required corrections